# OpenReview forum: "Quantifying the Salience of Geo-Cultural Values for Pluralistic Safety Alignment"
_ICML.cc/2026/Conference — ICML 2026 regular_

### Official Review · Reviewer_rqvU · 2026-03-10

**Soundness:** 3
**Presentation:** 3
**Significance:** 3
**Originality:** 3
**Overall Recommendation:** 4
**Confidence:** 4

**Summary:**

The authors examine several existing pluralistic alignment datasets for the effects of demographic and cultural factors. They find that both demographics and culture have predictive power on safety annotations, and that culture improves predictive power even after accounting for demographics. They propose a method for labeling items as culturally sensitive and find that, broadl, models perform poorly at classification tasks involving these culturally sensitive items.

**Compliance With Llm Reviewing Policy:**

Affirmed.

**Final Justification:**

The rebuttal fully addressed my main concerns regarding the need for qualitative discussion and more clarification of the cultural analysis methods.

**Key Questions For Authors:**

See above: [1] what evidence do you have that the proposed methods in sections 5-6 actually identify culturally sensitive items? [2] what steps were taken to check that the cultural quadrants actually represent distinct values expressed in the annotations?

**Limitations:**

yes

**Strengths And Weaknesses:**

Strengths:
- Analysis of a fairly large range of pluralistic alignment dataset functions as a useful meta-review on the relative importance of demographic and cultural factors.
- The paper is generally well-written, and examining the contribution of culture to disagreement in annotation, as opposed to demographics alone, is a large open question in this line of work.

Weaknesses:

In general, qualitative discussion would greatly help this paper. There is a lot of discussion of the importance of “culturally sensitive” items but little instantiation of what these actually look like—not a single example from any of the datasets is mentioned anywhere in the paper. What exactly are these culturally sensitive issues that, as the authors claim, traditional approaches do not capture? How do we know that the proposed methods are better at capturing them? In other words, what’s the face validity that the identified “culturally sensitive” items actually capture something culturally sensitive?

309: How do you know that something is purely cultural just because more than one demographic group is involved? E.g. suppose you have a dataset in which people ages 55-65 and 65+ pattern together. It might be a purely age-based trend, not based on some external cultural factor, but because more than one age group is involved, it would pass the validity filter.

An example would be helpful here to illustrate how the cultural quadrant and validity filtering works for a particular item.

I am a bit concerned about noise that can be introduced in the cultural measurement by going from a small set of individuals (3+) rating an item, mapping them to a set of geocultural values based solely on nationality, and then further coarsening this by lumping cultures by quadrants. I would expect this to result in odd constellations of values within a quadrant if, say, you have three people from South Korea and three people from Greece both meant to represent a single, cohesive geocultural group because their countries happen to be in the same quadrant. The multi-stage mapping from "individual" to "culture" without validation of the intermediate steps makes it difficult to judge the soundness of the analysis in sections 5 and 6.

---

> ### Author Rebuttal · Authors · 2026-03-31
>
> **Qualitative discussion**:  We agree the paper would benefit from expanded qualitative discussion, thank you for raising this. We will add 5 examples from each dataset to appendix. Some items surfaced by our method:
> - Example 1 (from D3): *Disgusting! How, many more decades will it take for the Roman Catholic Church to wise up*
>     - Q I (Secular-Rational, Self-Expression): 0/8 rated harmful
>     - Q IV (Traditional, Self-Expression): 5/5 harmful
> - Example 2 (from CREHate): *Best wishes Uganda . Fighting back against violent homosexuality  & the grooming*
>     - Q I: 11/15
>     - Q II (Secular-Rational, Survival): 1/5
> - Example 3 (from D3): *thoughts and prayers - yeah that'll save a lot of lives*
>     - Q I: 7/8
>     - Q III (Traditional, Survival): 1/9
> - Ex. 1 captures the Traditional vs. Secular-Rational divide on reverence for religious institutions. Ex. 2 illustrates the Self-Expression vs. Survival divide regarding pathologization of sexuality. Ex.3 shows not all divides are clearly explainable; further qualitative research is needed.
>
> **Methods for cultural sensitivity analysis**: To clarify, our metric quantifies the *empirical impact* of what would happen had *existing* datasets lacked geo-cultural diversity. Therefore, we operationalized "culturally sensitive" in a strictly measurable way: if excluding just one cultural value quadrant would flip the label to "uncontroversially safe" (i.e., it was deemed unsafe by exactly one quadrant).
> - To mitigate purely demographic confounders, we ensure more than one age, gender, and ethnicity is present within a quadrant. If *even one* of these is not satisfied, the quadrant was not considered. If less than 2 quadrant votes were present, the item was discarded.
> - Rater age was mapped to generational cohorts (genz, millennial, genx, boomer) which have value differences according to social science studies [1,2,3].
> - We use a Beta posterior to obtain more robust joint probability estimates given the low number of raters (see response to oUVC on cult. sens. items).
> - This is the most rigorous validity filtering possible with current dataset limitations. However, we acknowledge missing quadrants could lead to false positives or negatives (see resp. to oUVC). We will clarify the limitations & emphasize future datasets should stratify by both demographics & geo-cultural background given our results.
>
> **Comparison to traditional approaches**:
> - Past works have taken a top-down approach to defining culturally sensitive items (e.g. foods, social norms) [5]. We take a bottom-up approach: starting from people’s annotations, seeing where raters along cultural value dimensions disagree.
> - In Section 6.1, we do not claim that prior approaches do not capture culturally sensitive items. Rather, we compare the standard safe/unsafe classification task to the safe/sensitive task. While the safe-sensitive task is harder, we actually find fine-tuned LMs achieved relatively high recall (80%), suggesting potential in prioritizing data for expensive multicultural annotation.
>
> **Choice of geo-cultural attributes**:
> - *Country to values*: We focus on geo-cultural values, i.e. values based on geographical context, on the country level. This choice is based on (1) seminal works in psychology & sociology establishing country-level differences in cultural values, and (2) reducing the complexity of additional rater stratification in a theory-driven way. The IW axes are derived from a comprehensive longitudinal survey of values worldwide.
> - *Values to quadrants*: Prior work investigated value alignment on IW quadrant level [4]. A coarser view (quadrants) was necessary to ensure multiple raters per group per item for the Section 5 analysis. In a new analysis we confirmed improved fit with hierarchical linear modeling (see response to w1Tz).
> - *Rater to country*:  We were principled in our choice of the exact variable to represent geographic-association, e.g. nationality vs. country of birth, and will include details in Appendix.
>
> ## Questions
>
> (1) See above.
>
> (2) Section 4 provides evidence of cultural impact controlling for all available confounders. Building on this, Section 5 establishes an estimate of how reducing quadrant diversity would empirically impact safety ratings. These do not require inference of value impact beyond aggregate level. However, we can observe limited qualitative evidence that these disagreements align with I-W value dimensions (see above).
>
> [1] Dimock, Michael. Defining generations: Where Millennials end and Generation Z begins.
>
> [2] Twenge et. al Generational differences in young adults' life goals, concern for others, and civic orientation
>
> [3] Leijen et. al, Individual and generational value change in an adult population, a 12-year longitudinal panel study.
>
> [4] Zhang et. al, Cultivating Pluralism In Algorithmic Monoculture: The Community Alignment Dataset
>
> [5] Liu et. al, Culturally Aware and Adapted NLP: A Taxonomy and a Survey of the State of the Art

---

> > ### Author Rebuttal · Reviewer_rqvU · 2026-04-03
> >
> > Thanks, I think the proposed changes and clarifications will strengthen the paper, and have updated my score accordingly.

---

> > > ### Author Response · Authors · 2026-04-06
> > >
> > > Thank you for your acknowledgment of our rebuttal and for acknowledging all your concerns are resolved via our rebuttal including:
> > > - Provision of qualitative examples;
> > > - Clarification on the cultural sensitive item analysis;
> > > - Clarification on comparison to traditional approaches in surfacing culturally sensitive items;
> > > - Clarification on the rationale behind the choice of geo-cultural attributes and the mapping from annotator to culture.
> > >
> > > We are grateful for your time and the feedback you have provided throughout this process.
> > >
> > > We are particularly encouraged by **your positive assessment of our work's strengths** in your initial review:
> > > - noted that our study is large-scale (also mentioned by w1TZ, oUVC) and the usefulness of our meta-review on the relative importance of demographic and cultural factors  (also mentioned by w1TZ, oUVC);
> > > - characterized our paper as well-written;
> > > - discussed our work as addressing a large open question (also mentioned by BWL3).
> > >
> > > We recall our paper’s contributions to the field of pluralistic safety alignment:
> > > - A large-scale meta-analysis of geo-cultural representation in current safety datasets;
> > > - Statistical evidence of cultural salience beyond demographics;
> > > - Empirical quantification of the importance of geo-cultural diversity;
> > > - Opportunities and limits of LLM-as-a-judge approaches;
> > > - Practical recommendations for future data collection based on our study.
> > >
> > > Thus, given your positive evaluation of our paper's research question, results, and writing combined with the rebuttal fully addressing your concerns, we were optimistic this would be sufficient to raise the score to “Accept”.

---

### Official Review · Reviewer_BWL3 · 2026-03-10

**Soundness:** 3
**Presentation:** 3
**Significance:** 3
**Originality:** 3
**Overall Recommendation:** 5
**Confidence:** 4

**Summary:**

The authors aim to analyze a central area in AI alignment: how geo-cultural differences affect judgments about whether AI outputs are safe or harmful. The paper begins with a meta-analysis of existing safety datasets and shows that very few of them record geo-cultural information about annotators, even though such variation may matter for evaluating safety.

To study this more systematically, the authors use cultural value dimensions derived from the World Values Survey and apply hierarchical modeling across several datasets. They find that geo-cultural background significantly predicts safety ratings beyond standard demographic variables. The paper also introduces a “cultural sensitivity score” to identify items where cultural differences meaningfully affect safety judgments and estimates that roughly 10% of items in current datasets could be cultural blind spots if rater pools lack cultural diversity.

Finally, the authors test whether language models can detect culturally sensitive items or emulate culturally specific judgments. They find that models can sometimes identify sensitive items but are not reliable at reproducing the judgments of raters from different cultural backgrounds. Overall, this manuscript's central contribution concerns providing empirical evidence and tools for analyzing geo-cultural variation in safety annotation and highlighting the need for culturally pluralistic evaluation in AI alignment.

**Compliance With Llm Reviewing Policy:**

Affirmed.

**Final Justification:**

I was already ready to accept the paper (before rebuttal) and am maintaining my rating of Accept.

**Key Questions For Authors:**

Q1. Concrete examples of culturally sensitive items. The paper provides strong statistical evidence that safety judgments vary across cultural groups, but the phenomenon remains somewhat abstract. Could the authors include one or two concrete examples of items where raters from different cultural quadrants disagreed in their safety judgments? Even anecdotal examples would help readers better understand the practical implications of the proposed framework.

Q2. Choice and limitations of the cultural framework. The analysis relies on the Inglehart–Welzel cultural value framework derived from the World Values Survey. Could the authors elaborate on why this framework was chosen over alternative approaches to modeling cultural variation? In particular, it would be helpful to briefly discuss how this framework compares to earlier approaches such as Hofstede’s cultural dimensions and why it is preferable in this context.

Q3. Implications for dataset design. The paper shows that roughly 10% of items in existing datasets may represent cultural blind spots when rater pools lack geo-cultural diversity. Based on your findings, what practical guidance can you offer for dataset creators? For example, how many raters per cultural group (or quadrant) might be needed to reliably detect culturally sensitive items?

Q4. Scope of cultural variation captured by the model. The analysis uses country-level cultural clusters as proxies for cultural values. How should readers interpret these results in light of potential within-country variation (e.g., regional, linguistic, or subcultural differences)? Do the authors expect the framework to extend naturally to finer-grained cultural groupings if such data were available?

**Limitations:**

Yes

**Strengths And Weaknesses:**

The paper has the following strengths:

S1. Important and underexplored problem. The paper addresses a meaningful gap in current AI safety and alignment practice: the lack of geo-cultural diversity in safety annotations. As AI systems are deployed globally, understanding how safety judgments vary across cultures is an important problem, and the paper makes a strong case that current datasets rarely capture this variation.

S2. Careful operationalization of a difficult concept. Cultural variation is inherently difficult to operationalize without becoming overly reductionist. The authors make a thoughtful attempt to operationalize geo-cultural values using an established social science framework and are appropriately cautious about the limitations of this approach. The methodology reflects due diligence in tackling a complex construct.

S3. Methodologically sound empirical analysis. The use of hierarchical (multilevel) models to separate annotator and item effects is appropriate and strengthens the analysis. The comparison between demographic-only and culture-aware models provides a clear empirical test of the paper’s main hypothesis.

S4. Concrete empirical contributions. The dataset audit, the proposed cultural sensitivity score, and the estimate of cultural blind spots in safety datasets provide tangible outputs that may be useful for future dataset design and safety evaluation work.

S5. Insightful negative results about LLM automation. The experiments showing that language models cannot reliably emulate culturally specific safety judgments are valuable and highlight limitations of LLM-as-a-judge approaches in pluralistic alignment settings.

There are also a couple of weaknesses, namely:

W1. Cultural proxies remain coarse. The analysis relies on country-level cultural value clusters derived from large surveys. While this is a reasonable starting point, such proxies inevitably simplify complex cultural variation and may mask substantial within-country heterogeneity.

W2. Limited discussion of alternative cultural frameworks. The paper uses the World Values Survey / Inglehart–Welzel framework, which is reasonable, but the broader literature on cultural measurement (including critiques of earlier frameworks such as Hofstede’s dimensions) could be discussed more explicitly to situate this choice.

W3. Limited validation with domain experts. The pathologist-based validation is interesting but quite limited in scope. Additional examples or discussion of how expert validation might scale would help clarify the practical implications of the findings.

---

> ### Author Rebuttal · Authors · 2026-03-31
>
> We thank the reviewer for their comments!
>
> > W1. Cultural proxies remain coarse. The analysis relies on country-level cultural value clusters derived from large surveys. While this is a reasonable starting point, such proxies inevitably simplify complex cultural variation and may mask substantial within-country heterogeneity.
>
> We agree that future work should investigate within-country cultural variation. Unfortunately, currently no safety datasets report such information, and as our study shows, few attempt even country-level diversification. Our study shows even at this coarse level, geo-cultural diversification yields benefits beyond demographics, which has not been shown before.
>
> > W2. Limited discussion of alternative cultural frameworks. The paper uses the World Values Survey / Inglehart–Welzel framework, which is reasonable, but the broader literature on cultural measurement (including critiques of earlier frameworks such as Hofstede’s dimensions) could be discussed more explicitly to situate this choice.
>
> We will include a more specific critique of prior frameworks. For example, Hofstede’s dimensions are based on a survey of IBM employees rather than longitudinal surveys of representative country samples.
>
> > W3. Limited validation with domain experts. The pathologist-based validation is interesting but quite limited in scope. Additional examples or discussion of how expert validation might scale would help clarify the practical implications of the findings.
>
> Our goal was to understand how cultural variation affects human safety judgements. We would expect this variation to be reduced for expert annotators, since they might follow pre-defined rubrics rather than reflect their perceptions of safety. Instead, we focus on bottom-up surfacing of disagreement among lay people. See discussion on annotator disagreement with reviewer oUVC.
>
> > Q1. Concrete examples of culturally sensitive items. The paper provides strong statistical evidence that safety judgments vary across cultural groups, but the phenomenon remains somewhat abstract. Could the authors include one or two concrete examples of items where raters from different cultural quadrants disagreed in their safety judgments? Even anecdotal examples would help readers better understand the practical implications of the proposed framework.
>
> We will provide examples in Appendix, see response on qualitative discussion to reviewer rqvU.
>
> > Q2. Choice and limitations of the cultural framework. The analysis relies on the Inglehart–Welzel cultural value framework derived from the World Values Survey. Could the authors elaborate on why this framework was chosen over alternative approaches to modeling cultural variation? In particular, it would be helpful to briefly discuss how this framework compares to earlier approaches such as Hofstede’s cultural dimensions and why it is preferable in this context.
>
> We focus on geo-cultural values, i.e. values based on geographical context, on the country level. This choice is based on (1) seminal works in psychology & sociology establishing country-level differences in cultural values, and (2) reducing the complexity of additional rater stratification in a theory-driven way. Specifically, the IW axes are derived from a comprehensive longitudinal survey of values worldwide, and country clusters are based on geographical & cultural (including religious) contexts.
>
> > Q3. Implications for dataset design. The paper shows that roughly 10% of items in existing datasets may represent cultural blind spots when rater pools lack geo-cultural diversity. Based on your findings, what practical guidance can you offer for dataset creators? For example, how many raters per cultural group (or quadrant) might be needed to reliably detect culturally sensitive items?
>
> Our findings indicate that 1) fine-tuned LMs could be used to prioritize items for diverse cultural annotation; 2) both demographic and geo-cultural diversity have significant impact on safety rating valuation. In light of these findings, dataset creators should recruit raters across all geo-cultural clusters. See also response to oUVC, Question 4.
>
> > Q4. Scope of cultural variation captured by the model. The analysis uses country-level cultural clusters as proxies for cultural values. How should readers interpret these results in light of potential within-country variation (e.g., regional, linguistic, or subcultural differences)? Do the authors expect the framework to extend naturally to finer-grained cultural groupings if such data were available?
>
> We acknowledge lack of intra-country nuance as a limitation (L397-410). Currently, safety datasets do not provide intra-cultural variation data. If such datasets were to be collected, the linear modeling / cultural sensitivity framework could easily be extended, by taking, for example, sub-cultural units as levels of variation instead of cultural clusters or quadrants.

---

> > ### Author Rebuttal · Reviewer_BWL3 · 2026-04-01
> >
> > I was already prepared to accept the paper, thank you for the additional input.

---

### Official Review · Reviewer_w1Tz · 2026-03-13

**Soundness:** 3
**Presentation:** 3
**Significance:** 3
**Originality:** 3
**Overall Recommendation:** 4
**Confidence:** 3

**Summary:**

This work conducts a meta-analysis of safety datasets for language models, focusing on the importance of geo-cultural values. The authors review 1,062 existing safety datasets, showing that geo-cultural information is missing in most cases despite its importance, as reflected in its predictive power for human safety annotations. The paper further shows that there are culturally sensitive items for which differences in geo-cultural values could affect real-world deployment when such factors are not taken into account. By fine-tuning language models on both safe-unsafe classification and safe–sensitive classification tasks, the authors demonstrate that current models have limited ability to identify controversial items and to simulate the safety judgments of human annotators from diverse cultural backgrounds.

**Compliance With Llm Reviewing Policy:**

Affirmed.

**Final Justification:**

The authors have fully addressed my concerns regarding the further justification for the heavy reliance on the Inglehart-Welzel cultural map and presentation. However, the central limitation (the reductionist nature of division into cultural clusters or quadrants) remains as acknowledged in the limitations section, so I maintain my score of 4.

**Key Questions For Authors:**

Please see weaknesses W1, W2, W3.

**Limitations:**

Yes

**Strengths And Weaknesses:**

Strength
- This work conducts a large-scale review of 1062 existing safety datasets, highlighting the clear gap between the importance of geo-cultural values and their limited consideration in existing datasets.
- The overall structure of the analysis is well-designed and conducted systematically.
- The limitations section discusses caveats in interpreting the empirical results by specifying the limitations in empirical choices, helps readers better understand what the findings imply.

Weaknesses
- W1) This study relies heavily on Inglehart-Welzel cultural map throughout the analysis in Section 5,6. It represents cultures along only two dimensions, and they are further simplified into cultural quadrants. Although some of these limitations are acknowledged in the paper, it would be helpful to provide a justification for this design choice and to further elaborate on why this framework is still meaningful for the analysis.
- W2) When evaluating cultural sensitivity with LLMs (Sec. 6.2), the paper provides cultural quadrants instead of actual country names. This setup may lower LLMs performance, potentially making the task more difficult than it would be in realistic settings.
- W3) Although the study primarily focuses on safety, the title includes 'alignment' without mentioning 'safety'. This may be confusing for some readers about the scope of the paper.
- The fonts in the figure 1 are to small to read.

---

> ### Author Rebuttal · Authors · 2026-03-31
>
> We thank the reviewer for their comments!
>
> **Reliance on Inglehart-Welzel cultural map in Section 5,6**:
> - We focus on geo-cultural values, i.e. values based on geographical context, on the country level. This choice is based on (1) seminal works in psychology & sociology [1-3] establishing country-level differences in cultural values, and (2) reducing the complexity of additional rater stratification in a theory-driven way. The IW axes are derived from a comprehensive longitudinal survey of values worldwide, and country clusters are based on geographical & cultural (including religious) contexts. We note Section 4 also relies on the IW dimensions.
>
> - Prior work investigated value alignment on IW quadrant level [4]. A coarser view (quadrants) was necessary to ensure multiple raters per group per item for the Section 5 analysis. Below, we provide new analysis via hierarchical linear modeling that demonstrates that both the cultural value axes as well as cultural quadrants are predictive of rater safety perception, confirming their validity for sections 5 & 6:
>
> | | **Value Axes (Traditional--Secular-Rational * Survival--Self-Expression)** | | | **Value Quadrants** | | |
> | :--- | :--- | :--- | :--- | :--- | :--- | :--- |
> | **Dataset** | **p** | **Δ AIC** | **% Δσ²** | **p** | **Δ AIC** | **% Δσ²** |
> | DIVE | < 0.001* | -13.98 | -2.99 | < 0.001* | -10.69 | -2.42 |
> | CulturalFrames | < 0.001* | -29.88 | -13.63 | 0.004* | -7.05 | -1.92 |
> | PRISM | 0.698 | 4.57 | 0.13 | 0.734 | 4.72 | 0.17 |
> | DICES-990 | 0.004* | -6.23 | -7.61 | 0.004* | -6.23 | -7.61 |
> | NLPos | 0.115 | 0.06 | -1.92 | 0.186 | 1.19 | -1.68 |
> | D3 | < 0.001* | -119.96 | -3.23 | < 0.001* | -208.59 | -5.52 |
> | CREHate | 0.004* | -7.14 | -2.35 | 0.002* | -8.08 | -2.34 |
> | Severity | < 0.001* | -46.43 | -3.59 | < 0.001* | -40.53 | -3.12 |
>
> **Country names instead of quadrants in Section 6.2**:
> - The goal is to estimate the LLM’s ability to emulate quadrant-level (Traditional–Secular-Rational and Survival–Self-Expression), not country-level ratings. Given a country name, the model would have to rely on implicit knowledge rather than explicit value descriptions. For example, given South Korea, the LLM would have to use implicit knowledge to derive that the country has more secular than traditional and more survival than self-expression values instead of just having those value definitions directly.
> - Note that there are multiple countries per quadrant. On a country level there can be very few raters – sometimes 1 or 2 rating a small number of items, which is not enough to fine-tune or evaluate a model. Finally, emulating specific country judgments would make the classification more difficult for the model rather than improve performance: currently, the set up is 4-label classification, which is much easier than e.g. 60-label (predicting safe/unsafe for each country).
>
> **Safety in title**: We will update the title to specify safety alignment.
>
> **Figure 1 too small**: We have redesigned the Figure to increase the readability and better convey the cultural zones on the Inglehart-Welzel map. The updated figure is available here: https://limewire.com/d/Tn5aY#9QFmZxdztO
> Caption: Mapping the geo-cultural diversity of raters in 8 safety datasets on the Inglehart-Welzel Cultural Map of the World. Dots are centroids of each cluster, size of the dot is how many raters are present. Underneath each cluster name, top 3 countries by annotator count and the number raters in that cluster are listed.
>
> ## References
>
> [1] Inglehart et al, Mapping global values
>
> [2] Hofstede, Culture's consequences: Comparing values, behaviors, institutions, and organizations across nations
>
> [3] Gelfand et. al, Differences Between Tight and Loose Cultures: A 33-Nation Study
>
> [4] Zhang et. al, “Cultivating Pluralism In Algorithmic Monoculture: The Community Alignment Dataset”

---

> > ### Author Rebuttal · Reviewer_w1Tz · 2026-04-03
> >
> > I thank the authors for clarifying my questions and providing additional analysis. I appreciate the updated figure, which is clearer than the one in the paper. I will maintain my positive score.

---

> > > ### Author Response · Authors · 2026-04-04
> > >
> > > Thank you for your acknowledgment of our rebuttal and for acknowledging all your concerns are resolved via our rebuttal which included:
> > > - Clarification regarding reliance on the I-W map use of quadrants instead of country names in 6.2;
> > > - Additional analysis that shows quadrant-level cultural signal is predictive of safety ratings and hence provides principled - support for using quadrant-level information for analysis in Section 5 and 6;
> > > - The updated figure with increased readability and clarity.
> > >
> > > We are particularly encouraged by **your positive assessment of our work's strengths** in your initial review:
> > > - noted that our study is large-scale (also mentioned by oUVC, rqvU) and highlights the gap between the importance of geo-cultural representation (also mentioned by oUVC) and its lack in existing work;
> > > - characterized our analysis as well-designed and systematic (also mentioned by BLW3);
> > > - mentioned the strength of our limitations section that clearly outlines empirical choices made in this work and helps understand the implications of our findings.
> > >
> > > We recall our paper’s contributions to the field of pluralistic safety alignment:
> > > - A large-scale meta-analysis of geo-cultural representation in current safety datasets;
> > > - Statistical evidence of cultural salience beyond demographics;
> > > - Empirical quantification of the importance of geo-cultural diversity;
> > > - Opportunities and limits of LLM-as-a-Judge approaches;
> > > - Practical recommendations for future data collection based on our study.
> > >
> > > We are grateful for your time and the feedback you have provided throughout this process. Given your positive evaluation of our paper's methodology, novelty, and results, combined with the rebuttal fully addressing all your concerns, we were optimistic this would be sufficient to raise the score to “Accept”.

---

### Official Review · Reviewer_oUVC · 2026-03-15

**Soundness:** 3
**Presentation:** 3
**Significance:** 3
**Originality:** 3
**Overall Recommendation:** 4
**Confidence:** 3

**Summary:**

The current LLM safety evaluations consider diverse datasets, giving the impression that they cover global raters from different cultures however, the evaluations actually belong to culturally narrow groups. The paper argues that age or gender or ethnicity attributes are not enough, but geo-cultural background matters. This leads to harm for the missing cultural group. The paper finds that only 9 out of 1062 datasets report both demographic and geo-cultural information. They validate their hypothesis by fitting hierarchical models showing that culture predicts safety ratings beyond standard demographics. For downstream applications, they show that LLM-Judge can flag culturally sensitive items for review, but it cannot simulate the geo-cultural persona.

**Compliance With Llm Reviewing Policy:**

Affirmed.

**Final Justification:**

As mentioned, my concerns were answered, and I have increased the score from 3-> 4.

**Key Questions For Authors:**

Please look at the weakness, here are some of the other questions:

- How do the authors define culture?

- After controlling for culture and demographics, do standard inter-rater agreement metrics (e.g., κ) improve or remain similarly low? How should readers interpret cultural effects in datasets where baseline annotator agreement is extremely low?

- Could some of the disagreement attributed to cultural differences instead reflect differences in task interpretation or dataset design?

- What types of data collection or annotation improvements would most help LLMs better model culturally grounded safety judgments in the future?

- How would cross-data generalisation work? This is the realistic scenario where LLM trained on properierty datasets are then being used in the wild by humans.

**Limitations:**

Yes

**Strengths And Weaknesses:**

# Strengths

- Very interesting and relevant research question. With LLM deployment beyond a single single/culture, it is important to shed a light on the datasets that are being used for safety alignment.

- Consider meta analysis of 1000+ datasets

- Strong evidence that adding culture on top of demographics improves fit.

# Weakness

- I struggle to find the correct definition of "culture". The paper operationalizes culture using country-level placement on the Inglehart–Welzel map, then often collapses further into quadrants. Since this paper has broader implications, I would suggest defining culture int he main paper more concretely. Atm I think it is using country level proxy but then I am not sure if this is truly "culture". Some countries could have multiple religions with very different cultures.

- I am unsure about the other datasets but DICES has an extremely low agreement rate between the annotators (< 0.2). The authors aim to move away from Cohen's kappa, but I think it is a standard metric used by dataset annotators. After reading the paper, I am still unsure (although I think it might increase slightly) if K is still low between the annotators. If it is still low then I am not sure if the claims can be as strong.

- Building on the prev one, some dataset/annotation papers [1] show that simply giving better annotation guidelines can result in higher agreement rate.

- Some bold claims seem a bit fragile. They define a culturally sensitive item as one where exactly one quadrant rates it unsafe, then estimate those probabilities using sparse votes plus a thresholding rule and a Beta posterior. But missing quadrants would not also have flagged the item.

- There may be confounders in the paper which are not accounted for - annotator proficiency/expertise.

- While it is out of scope for the rebuttal but such a paper would benefit from human study. Could the authors conduct a human study by controlling for confounders and using samples from these datasets to annotate and assess differences?


[1] Blandón, María Andrea Cruz, et al. "Memerag: A multilingual end-to-end meta-evaluation benchmark for retrieval augmented generation." Proceedings of the 63rd Annual Meeting of the Association for Computational Linguistics (Volume 1: Long Papers). 2025.

---

> ### Author Rebuttal · Authors · 2026-03-31
>
> **Defining culture**:
> Thank you for raising this, we will clarify the exact operationalization of culture and our reasoning (expanding on L190-209).
>
> We focus on geo-cultural values, i.e. values based on geographical context, on the country level. This choice is based on (1) seminal works in psychology & sociology [1-3] establishing country-level differences in cultural values, and (2) reducing the complexity of additional rater stratification in a theory-driven way. The IW axes are derived from a comprehensive longitudinal survey of values worldwide, and country clusters are based on geographical & cultural (including religious) contexts.
>
> We acknowledge this has limitations, including lack of intra-country nuance (L397-410). Yet our meta-analysis shows that the vast majority of safety datasets do not consider even country-level variation (L88-91), establishing our work as a critical first step.
>
> **Annotator disagreement**: Penalizing low IAA assumes all disagreement is noise. However, extensive research in the field of crowdsourced annotation has established it as a meaningful signal of variation in opinion [4-6].
> - Low IAA does not invalidate our findings: disagreement could stem from e.g. having multiple ethnicities or genders in the same group. Given low sample replication from the same demographic & cultural groups, we turn to hierarchical linear models that rigorously account for the confounders (rater subjectivity, item difficulty, etc.). They show that cultural clusters explain rater responses even after including demographics ($p<0.05$ in 6 datasets, Table 3).
> - While stricter rating guidelines increase IAA, they also suppress individual value-based variation, which is the subject of our study ([4-6]).
> - All datasets included in the study were peer-reviewed. We rely on them for basic quality control such as removing annotators that were flagged as unreliable.
>
> **Culturally sensitive items**: To clarify, our metric quantifies the *empirical impact* of what would happen had *existing* datasets lacked geo-cultural diversity. We agree we cannot make inferences about the missing quadrants (both false positives and negatives are possible). However, our analysis provides a rough statistical estimate based on the data available. We will clarify this in limitations.
> - We use a Beta posterior to reduce the number of false positives (estimates below and above 0.5 are pulled to 0.5, yielding a lower estimate of the joint probability).
> - $T_u$ for Likert scores is set to one level above “totally safe”, mirroring common zero-tolerance safety policies. $S_{iq}$ threshold is set to $>0.5$ enforcing a "more likely than not" standard.
> - We believe this is the best estimate achievable given the data constraints, but are happy to take any suggestions from the reviewers.
>
> **Human study**: We will expand our future work discussion to advocate for a high-powered human subject study of geo-cultural factors by following our practical recommendations.
>
> ## Questions:
>
> Q1-Q2: see above.
>
> Q3: Task interpretation differences often stem from task subjectivity wherein raters are enabled to/asked to bring in their values and lived experiences [4-6]. In this work we study the role of geo-cultural values in rater responses and statistically quantify their importance.
>
> Q4: We provide takeaways for future dataset design addressing practical implications of our findings in L421-436. To elaborate:
> - Include proxies for cultural values in addition to basic demographics (age, gender, race) – at least country of longest residence, birth, or self-identified cultural affiliation.
> - Instead of sampling geographically exhaustively, consider representation across established theory-driven cultural value dimensions (e.g., Traditional vs. Secular-Rational, Survival vs. Self-Expression).
> - In constrained annotation budget settings, fine-tuned LLM classifiers could be used to prioritize items for geo-culturally diverse annotation.
>
> Q5: We are not exactly sure what cross-data generalization means here. For models described in the paper, the use is meant for model providers for data annotation rather than humans in the wild. For models fine-tuned on safe-sensitive task (Sec 6.1), we find a moderate degradation in cross-domain performance: the model fine-tuned on D3 (71.2 F1 on test set) achieves 66.7 F1 on DICES and 63.7 F1 on CREHate, suggesting possible out-of-domain use with caution.
>
> --
>
> [1] Inglehart et al, Mapping global values
>
> [2] Hofstede, Culture's consequences: Comparing values, behaviors, institutions, and organizations across nations
>
> [3] Gelfand et. al, Differences Between Tight and Loose Cultures: A 33-Nation Study
>
> [4] Aroyo, et. al. Truth is a lie: Crowd truth and the seven myths of human annotation.
>
> [5] Denton et. al, Whose Ground Truth? Accounting for Individual and Collective Identities Underlying Dataset Annotation
>
> [6] Plank et. al, The “Problem” of Human Label Variation: On Ground Truth in Data, Modeling and Evaluation

---

> > ### Author Rebuttal · Reviewer_oUVC · 2026-04-01
> >
> > I would like to thank the authors for their rebuttal. My answers are resolved, I have increased my score to 4 as the paper tackles an overlooked problem.

---

> > > ### Author Response · Authors · 2026-04-06
> > >
> > > Thank you for your acknowledgment of our rebuttal and for acknowledging all your concerns are resolved via our rebuttal including:
> > > - Definition of culture and operationalization through the I-W map;
> > > - Clarification on the strengths of our analysis compared to IAA-based approaches;
> > > - Clarification on the cultural sensitive item analysis.
> > >
> > > We are grateful for your time and the feedback you have provided throughout this process.
> > >
> > > We are particularly encouraged by **your positive assessment of our work's strengths** in the initial review.
> > > - Relevance of the research question (also mentioned by BWL3, rqvU)
> > > - Large-scale of the meta-analysis (also mentioned by W1TZ, rqvU)
> > > - Evidence of the importance of geo-cultural representation (also mentioned by w1Tz)
> > >
> > > We recall our paper’s contributions to the field of pluralistic safety alignment:
> > > - A large-scale meta-analysis of geo-cultural representation in current safety datasets;
> > > - Statistical evidence of cultural salience beyond demographics;
> > > - Empirical quantification of the importance of geo-cultural diversity;
> > > - Opportunities and limits of LLM-as-a-judge approaches;
> > > - Practical recommendations for future data collection based on our study.
> > >
> > > Thus, given your positive evaluation of our paper's research question and results, combined with the rebuttal fully addressing your concerns, we were optimistic this would be sufficient to raise the score to “Accept”.

---

### Decision · Program_Chairs · 2026-04-30

**Decision:**

Accept (regular)

**Comment:**

The authors conduct a large-scale audit of safety annotation datasets and show, through hierarchical linear modeling, that geo-cultural cluster membership explains annotator disagreement beyond standard demographic variables, motivating more pluralistic approaches to safety evaluation. Reviewers agreed the problem is important and found the meta-analysis a useful community contribution; methodological concerns about the coarseness of the Inglehart-Welzel quadrant operationalization and modest effect sizes were acknowledged and addressed by the authors and do not substantially undercut the core argument. I recommend accept.